# MoIN: Mixture of Introvert Experts to Upcycle an LLM

## Abstract

The goal of this paper is to improve (upcycle) an existing large language model without the prohibitive requirements of continued pre-training of the full-model. The idea is to split the pre-training data into semantically relevant groups and train an expert on each subset. An expert takes the form of a lightweight adapter added on the top of a frozen base model. During inference, an incoming query is first routed to the most relevant expert which is then loaded onto the base model for the forward pass. Unlike typical Mixture of Experts (MoE) models, the experts in our method do not work with other experts for a single query. Hence, we dub them "introvert" experts. Freezing the base model and keeping the experts as lightweight adapters allows extreme parallelism during training and inference. Training of all experts can be done in parallel without any communication channels between them. Similarly, the inference can also be heavily parallelized by distributing experts on different GPUs and routing each request to the GPU containing its relevant expert. We implement a proof-of-concept version of this method and show the validity of our approach.

## 1 Introduction

As language models continue to grow in size, it is important to consider whether all parameters are necessary for every request. For instance, the same parameters might not be ideal for answering questions about SQL compared to those about car seats. Approaches like fine-tuning, early exit, and mixture of experts (MoE) have been proposed to address this issue, each with its own trade-offs. Fine-tuning requires meticulous data curation and optimization, while early exit and MoE add complexity to the model's forward pass, posing engineering challenges. Most relevant to our work is the Mixture of Experts (MoE) approach, popularized by the Mixtral 8x7B model (Jiang et al., 2024). Existing experts operate at the token level and not sequence level, thus lacking specialization in specific domains [1] (Jiang et al., 2024). Additionally, since routing varies by token, all experts must reside in GPU memory during both training and inference, making it difficult to scale to a large number of experts. To address these limitations, we focus on the concept of query-level or sequence-level experts, where each query is routed to a single expert. This design minimizes communication between experts, which can be distributed across different GPU nodes. We term these "introvert" experts due to their reduced interaction with one another.

Implementing sequence-level experts offers several practical advantages. It creates a modular architecture where a single generalist base model is complemented by multiple small experts that specialize in niche topics. While fine-tuning shares a similar motivation, the number of experts is often constrained by the need for curated datasets. In contrast, because the experts operate on independent domains, they can be trained in parallel, enhancing flexibility in resource utilization. Moreover, these experts can be retrained as data evolves, allowing for the easy integration of new experts as fresh data becomes available. This capability positions LLMs as continual learners with minimal risk of forgetting prior knowledge.

We present an intuitive and straightforward proof-of-concept for creating such experts. Given the high cost of pre-training, we focus on model upcycling, transforming an existing dense model into a variant of mixture of experts by introducing new parameters and additional training. We begin

---

[1] https://mixtral-moe-vis-d726c4a10ef5.herokuapp.com

with a partially trained dense model, freeze its parameters, and train new experts on top of it. We believe any LLM adapter module can serve as an expert; however, we utilize the popular LoRA adapter (Hu et al., 2022) in our experiments. To train an expert, we cluster pre-training data points into semantically relevant groups. During inference, a query is routed to its most relevant cluster, and the corresponding adapter is employed in the forward pass. We utilize a dataset containing 500 billion tokens and experiment with 500 and 5,000 expert LoRA adapters.

The main advantage of our method is that the training time resource requirements are greatly reduced compared to dense model training. The resource savings happen on two fronts: First, GPU memory needed to train each expert is reduced since the base model is frozen and only LoRA weights are trained ($\approx 10\%$ of the model weights). Second, there's no need to have a fast network channel between two expert training runs since the experts are independent of each other. In extreme, all experts can be trained in the time it takes to train the expert with the largest training data subset. Then, we can use many affordable GPUs instead of a few expensive ones. It is also possible to train the experts on GPUs of different architectures and speeds based on availability and price. Full-model training speed is dictated by the speed of the slowest GPU and is thus less efficient than our approach.

Given the large number of adapters generated by our method, efficiently serving them can be a problem. However, there are open source libraries like LoRAX [2] based on Punica kernels (Chen et al., 2024) that reduce the amortized cost of running multiple adapters in a single batch. A downside of our approach is the linear increase in the storage memory with increase in the number of topics. However, this can be mitigated when the framework is scaled to simultaneously serve a large number of users. Further, it is possible to cache the most frequently routed adapters on the GPU memory, thereby eliminating the latency of moving them from CPU memory to GPU memory. It is also worth noting that the bandwidth between CPU memory and GPU memory is improving each day with newer GH200 chips supporting speeds of up to 900 GB/s [3]. This makes the CPU memory to GPU memory latency hit significantly less.

To summarize, our contributions are as follows. First, we propose framework called mixture of introverts for efficiently training and serving sequence level experts. Second, as a proof-of-concept, we apply our idea to the task of upcycling an LLM and show that our model is comparable in perplexity to a baseline model that is trained on 500B more tokens than our method.

## 2 RELATED WORKS

**Mixture of Experts**. Instead of activating all parameters for all inputs, conditional computing aims to activate only a subset of the parameters for each input (Jacobs et al., 1991; Bengio et al., 2013; Bengio, 2013). The goal is to increase the capacity of a model without increasing the compute of the model (Cho & Bengio, 2014). However, the idea didn't become widely known until the introduction of Mixture-of-Experts layer (Shazeer et al., 2017). This layer was used in training the Mixtral 8x7B model (Jiang et al., 2024) that further popularized this approach. Other variations of the MoE layer include (Lample et al., 2019; He, 2024; Rajbhandari et al., 2022; Dai et al., 2024). However, since the MoE creates experts in the MLP block of the transformer architecture, experts are created token-wise. When the model size increases and all experts cannot reside on a single GPU, complicated multi-GPU setups involving scatter and gather operations are needed at each MoE layer. Hence, this paper explores sequence-wise conditional computation / experts. This has the advantage of only needing to load weights relevant to a given query in the GPU while rest of the weights can reside in CPU RAM or even hard disk, significantly simplifying the architecture design and implementation.

The goal of this paper is to create topic-wise experts. Conversely, it is possible to identify parts of a dense model that only activate for certain topics. Essentially, discover instead of train experts. This was studied in (Dai et al., 2021).

**LoRAs for efficient distributed training of LLMs.** Recent works (Huh et al., 2024b; Lialin et al., 2024; Zhao et al., 2024) have shown the potential for a branch-train-merge distributed algorithm where low rank adapter parameters are branched, trained and then merged back into the base model

---

[2]https://github.com/predibase/lorax

[3]https://resources.nvidia.com/en-us-data-center-overview-mc/en-us-data-center-overview/grace-hopper-superchip-datasheet-partner

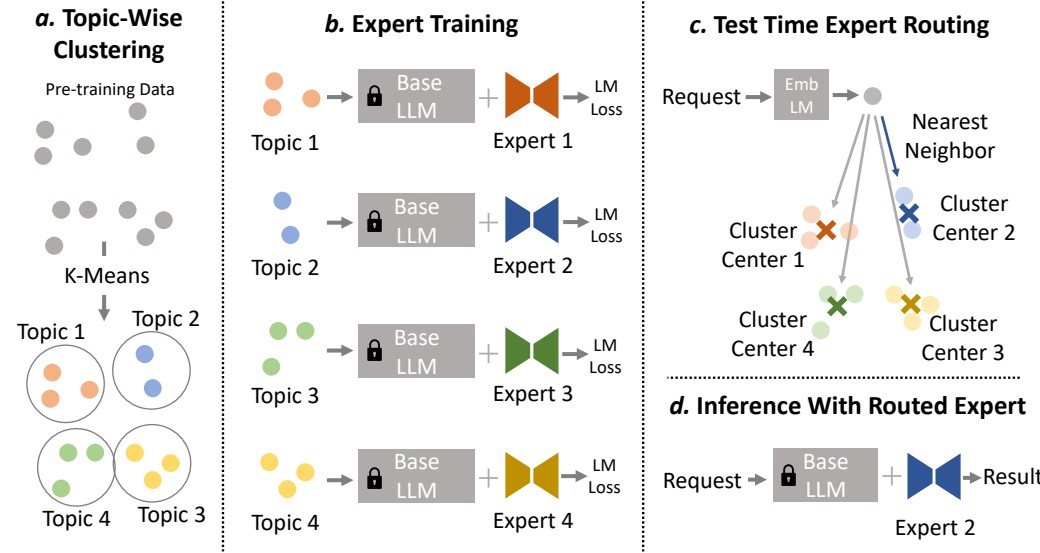

Figure 1: **Training and Inference with Topic-wise Experts.** Expert training involves two steps: (a) clustering the training data into semantically related subsets and (b) parallelly training a topic-wise expert model on each data subset. (c) At inference, the entire query is routed to the most appropriate expert using a simple nearest-neighbor search. (d) The expert is then loaded alongside the base model and the request is forwarded through the model.

weights. This reduces the number of parameters being trained which frees up GPU memory for a higher batch size resulting in better hardware utilization (Lialin et al., 2024). Another recent work (Zhao et al., 2024) uses low rank gradient projections to design a better optimizer. Similar to these works, our motivation is to reduce the cost of training a model, but we focus on upcycling where a pre-trained LLM is available. This allows us to completely parallelize the training without any communication between experts.

**LoRA based multi-task learning**. LoRA adapters have seen growing applications in the are of multi-task learning, owing to their lightweight nature. PEER (Parameter Efficient Expert Retrieval) (He, 2024) introduced LoRA based expert layer for augmenting/replacing feed forward layers. PaLoRA (Dimitriadis et al., 2024) uses task-specific low rank adapters for multi-objective optimization problems. LoRA the Explorer introduced in (Huh et al., 2024a) trained a neural network from scratch using averaged gradients from parallel LoRAs.

**Continual learning**. Since state of available information and human knowledge is dynamic, it is necessary to keep the LLMs up to date with these changes. This was investigated in (Jang et al., 2022). Hence, the aim of continual learning is to adapt the model to updated data without forgetting the previous knowledge (catastrophic forgetting) (Kirkpatrick et al., 2017). Since LLMs are supposed to operate across various domains, there is a need to improve specialized knowledge on new tasks without reducing significant performance on previously learned domains and tasks. On this end, (Ke et al., 2023) introduced DAS (Continual DA-pre-training of LMs with Soft-masking). (Jang et al., 2021) introduced continual knowledge learning, a method for updating temporal knowledge in LLMs, reducing forgetting while acquiring new information.

## 3 METHOD

Our key idea is to improve a (partially) pre-trained LLM by training topic-wise expert models atop it and dynamically routing the queries to the appropriate expert during inference. See Figure 1 for an illustration of the steps involved in our method.

### 3.1 TOPIC MODELING

In order to train the topic-wise experts, we need a dataset with topic annotations. For broad coverage of topics, we focus on using large-scale datasets typically used in LLM pre-training. Since these are usually web-scale text datasets without any topic annotations, we need to perform topic modeling before training the expert models.

Here, we consider two approaches for topic modeling. In both approaches, all documents in the training dataset are converted to embeddings using a pretrained sentence embedding model. In the first method, we employ UMAP (McInnes et al., 2018) to reduce the dimensionality of the document embedding and then perform clustering in the lower dimension space using HDBSCAN (McInnes et al., 2017). Topic-wise embeddings are then obtained by analysing the document belonging to each cluster. The topic embeddings are needed to route the queries to corresponding topics during inference. We employ this topic modeling approach in our model termed MoIN-500 where we split the dataset into 500 topics. We find that it is difficult to scale this approach to thousands of topics with limited resources. Thus, we consider a simpler alternative. In the second approach, we perform a simple K-means clustering directly in the document embedding space to generate cluster index for each training document and 'k' cluster centers as topic embeddings.

### 3.2 TRAINING OF TOPIC-WISE EXPERT MODELS

Given a set of topics, we train an expert model for each topic atop our base pretrained model. Since the number of models scales linearly with the number of topics, we need the number of trainable parameters in the expert models to be relatively small compared to the base model. Recently, parameter-efficient approaches (Hu et al., 2022; Koohpayegani et al.; Liu et al., 2024) have been very popular for model fine-tuning. Here, we consider them for the upcycling task. Specifically, we use LoRA (Hu et al., 2022) architecture for the experts. In LoRA, the base model is frozen and a few additional parameters are introduced for each linear layer in the network. The output of a linear layer is modified to be $y = Wx + W_b(W_a x)$ where $W \in \mathbb{R}^{k \times d}$ is the original weight matrix, $x$ is the input vector, $d$ is the dimensionality of input, $k$ is the dimensionality of output, and $W_b \in \mathbb{R}^{k \times r}$ and $W_a \in \mathbb{R}^{r \times d}$ are the trainable parameters. $W_b$ and $W_a$ are designed to be low rank matrices, i.e., $r \ll \min(d, k)$, thus reducing the number of newly added parameters. For every topic in the training dataset, we train an expert LoRA adapter using the documents assigned to the topic. Unlike typical LoRA fine-tuning on task-specific loss, the LoRA models here are trained using the standard autoreggressive language modeling loss (Radford, 2018) on subsets of the pretraining data. For each expert training run, only the weights of the expert are trained while the base model weights remain frozen. The key advantage over a standard pretraining is that all the experts can be trained independently, allowing for a great flexibility in the resources used for training.

### 3.3 MODEL INFERENCE

Once all experts have been trained, our setup consists of a large number of experts that must be served efficiently during inference. For each inference request, we need to find a topic-wise expert that can best serve it, and use the corresponding adapter during the forward pass through the model. We perform this 'query routing' in the following two steps: first, we embed the query using the same embedding model used to cluster the training dataset. Second, we perform nearest neighbor search over the topic embeddings using the query embedding. Since this routing step is an overhead specific to our method, the router model should be small enough to keep the inference latency feasible. To achieve this, we use a small document embedding model with just 20M parameters. Given the small size, it may be possible to host it without GPUs or to perform routing entirely on the client side. However, while the small size of the model helps reduce router latency it can also reduce the quality of the topic clustering. Router latency vs. clustering accuracy is an inherent trade-off in this method. In our current implementation, we choose faster inference over a more complex but better topic modeling.

## 4 EXPERIMENTS

**Implementation details:** For topic modeling, we use a small language model `all-MiniLM-L6-v2` with 20M parameters as the embedding model. We use two variants

of our method - a large scale experiment termed MoIN-5k with K-means based topic modeling and a smaller scale one termed MoIN-500 with HDBSCAN based modeling. In K-Means clustering to generate the topics, the number of clusters $k$ is set to 5000. Since some of the cluster sizes were extremely small, we retained only 4697 of the 5000 clusters and trained a LoRA for each of those 4697 topics. In MoIN-500 , the number of clusters is set to 500. Of them, we retain 387 experts and discard the rest since their training data is limited.

For LoRA training, we use the open-sourced TinyLlama (Zhang et al., 2024) model as the base network. TinyLlama uses the same architecture and tokenizer as Llama-2 (Touvron et al., 2023) but with a smaller parameter count of 1.1B. The model is trained for 3T tokens in total and additional intermediate checkpoints are open-sourced. To facilitate easier comparison with suitable baselines, we use the intermediate checkpoint of TinyLlama trained for two trillion tokens as the base model. The LoRA training follows the recipe from LitGPT (AI, 2023) and uses the official code from TinyLlama [4]. We use a single rank of 128 (token dim $d$ is 2048) for all the LoRAs and optimize them for one epoch on their corresponding topic data subsets. Following LitGPT, we use AdamW (Loshchilov & Hutter, 2019) optimizer and set $\beta_1 = 0.9$ and $\beta_2 = 0.95$. The learning rate is controlled using a cosine scheduler with a maximum lr of $4 \times 10^{-4}$ and a minimum lr of $4 \times 10^{-5}$. An effective batch size of 576 is used in optimizing all LoRAs. MoIN-5k is trained for 500B tokens in total while MoIN-500 is trained on just 50B tokens. Due to resource constraints, we limit the experiments to train just two sets of experts. These experiments serve as proof-of-concept of our approach. To perform the LoRA training, we use either Nvidia RTX-6000 or RTX-3090 GPUs with 48GB and 24GB memory respectively. This is an advantage of our method compared to other pretraining methods that we can use a mixture of different types of GPUs (whatever is available in our GPU cluster) since the LoRA training jobs are independent of each other.

**Datasets:** TinyLlama (Zhang et al., 2024) uses SlimPajama (Soboleva et al., 2023) and Star-Coder (Li et al., 2023) datasets for model pretraining. Even though we employ LoRA, our goal is to have a model that generalizes well on any downstream application. Thus we use the pretraining datasets for our LoRA/expert training. We use only the SlimPajama dataset for our LoRA training. It is a deduplicated and cleaned version of RedPajama dataset and contains 627B tokens. Following TinyLlama, we exclude the GitHub subset from the dataset. To enable comparison with TinyLlama models, we use approximately 500B tokens for training and the rest of the dataset for validation. Additionally, we perform a smaller scale experiment with a training set of just $50B$ tokens sampled randomly from the dataset.

**Evaluation:** We conduct downstream evaluation using the lm-eval-harness Gao et al. (2024) code and use six benchmark datasets for commonsense reasoning. The datasets include HellaSwag (Zellers et al., 2019), PIQA (Bisk et al., 2020), WinoGrande (Sakaguchi et al., 2021), OpenBookQA (Mihaylov et al., 2018), AI2 ARC easy and challenge (Clark et al., 2018) and BoolQ (Clark et al., 2019). We perform zero-shot evaluation on all downstream tasks.

**Baselines:** We primarily compare with different intermediate training checkpoints of the TinyLlama model. The TinyLlama model trained for 2T tokens (denoted as TinyLlama-2T in tables) is used as the frozen base model on top of which we train the LoRAs. Since we use nearly 500B tokens for training, our main comparison point is the TinyLlama model trained with equivalent number of tokens, that is, the 2.5T checkpoint. We also provide comparison with the final checkpoint of TinyLlama trained for 3T tokens.

### 4.1 RESULTS

**Pretraining:** Table 1 reports the perplexity of the baselines and MoIN-5k on the SlimPajama validation set. For evaluating MoIN-5k model, we first determine the topic for each document using our K-means based topic model and use the corresponding LoRA for perplexity calculation. MoIN-5k not only outperforms the equivalent TinyLlama-2.5T but also achieves comparable performance to TinyLlama-3T which is trained with 500B more tokens. Figure 2 depicts the perplexity for each LoRA model. Most of the models perform well with a perplexity lower than 10. For the underper-

---

[4]https://github.com/jzhang38/TinyLlama

Table 1: **Model perplexity on SlimPajama dataset.** We calculate the perplexity of the baseline and our LoRA based models on the validation set of SlimPajama dataset. As expected, higher amounts of training results in lower perplexity for the baseline methods. MoIN-5k trained on 2.5T tokens outperforms the corresponding baseline and is comparable to the model trained on 3T tokens.

| Model | Perplexity | Training Tokens |
|---|---|---|
| TinyLlama-2T | 9.209 | 2.0T |
| TinyLlama-2.5T | 8.260 | 2.5T |
| TinyLlama-3T | **8.137** | 3.0T |
| MoIN-5k | 8.167 | 2.5T |

Table 2: **Model performance on downstream tasks.** We perform zero-shot evaluation on seven benchmark datasets. MoIN-5k performs comparably to TinyLlama-2.5T and marginally outperforms it on average. Surprisingly, we observe that the baseline trained on 2.5T performs better than the one trained on 3T tokens on most datasets.

| Model | Train Tokens | HellaSwag | OBQA | WinoGrande | ARC_C | ARC_E | BoolQ | PIQA | Avg |
|---|---|---|---|---|---|---|---|---|---|
| TinyLlama-2T | 2.0T | 54.63 | 21.4 | 56.83 | 28.07 | 54.67 | **63.21** | 70.67 | 49.93 |
| TinyLlama-2.5T | + 0.5T | 58.96 | 24.2 | 58.72 | **31.91** | 56.77 | **63.21** | 73.06 | 52.40 |
| TinyLlama-3T | + 1.0T | **59.20** | 21.8 | **59.12** | 30.12 | 55.34 | 57.83 | **73.29** | 50.96 |
| MoIN-500 | + 0.05T | 56.07 | 24.8 | 58.48 | 28.84 | 54.80 | 62.78 | 72.03 | 51.12 |
| MoIN-5k | + 0.5T | 58.13 | **25.4** | 58.48 | 31.57 | 58.37 | 63.00 | 72.25 | **52.45** |
| MoIN-4697 | + 0.5T | 58.14 | 25.2 | 58.64 | **31.91** | 58.42 | 62.84 | 71.87 | 52.43 |

forming ones, it is possible to improve them by training them more and with additional data. Since all the LoRAs are trained and used independently, the performance of one will not affect the others.

**Downstream tasks:** Similar to perplexity evaluation, we first determine the topic for each document using our K-means based topic model and use the corresponding LoRA for calculating metrics for MoIN-5k . In MoIN-5k , if the LoRA model is absent for a given topic, just the base model is used. However, in MoIN-4697 , the query is always routed to one of the trained $4697$ LoRA models and the baseline alone is not used for any of the queries. The performance of MoIN-5k is comparable to that of TinyLlama-2.5T on nearly all the downstream tasks with MoIN-5k being marginally better on average. Surprisingly, TinyLlama-2.5T is better than TinyLlama-3T on five of the seven tasks.

For any given dataset, not all the LoRAs necessarily participate in the evaluation process. This is particularly true for datasets with very few total queries or for those focussed on a narrow set of topics. Table 3 shows the number of unique LoRAs used in the evaluation for each downstream task. We observe that usually only about $10\%$ of the LoRAs are used for a single task.

Table 3: **Number of unique LoRAs used in downstream tasks.** For evaluation on downstream tasks, each input query is routed to one of the LoRAs and the accuracy of the corresponding output is calculated. For a given dataset, not all the LoRAs necessarily participate. This is particularly true for datasets with very few total queries. We observe that usually only about $10\%$ of the LoRAs are used for the given task.

| | HellaSwag | OBQA | WinoGrande | ARC_C | ARC_E | BoolQ | PIQA |
|---|---|---|---|---|---|---|---|
| # Queries | 40145 | 2000 | 2534 | 4687 | 9496 | 6540 | 3676 |
| # Unique Loras | 1877 | 403 | 562 | 459 | 583 | 1235 | 692 |

**Topic modeling:** We explore two methods for topic modeling - one with 500 topics and the other with 5000 topics. For the smaller variant, we use HDBSCAN to cluster the topics and generate topic embedding by utilizing the documents belonging to each cluster. For the larger variant, we use a simple K-means clustering of all documents and use the cluster centroid as the topic embedding. For both methods, a nearest neighbour search is performed using the query and topic embeddings to

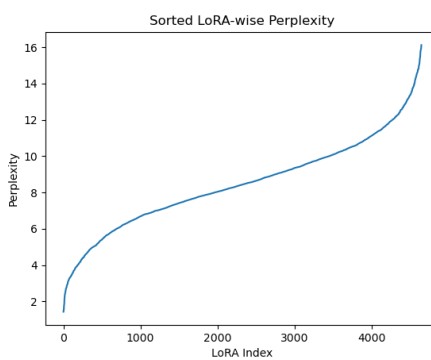
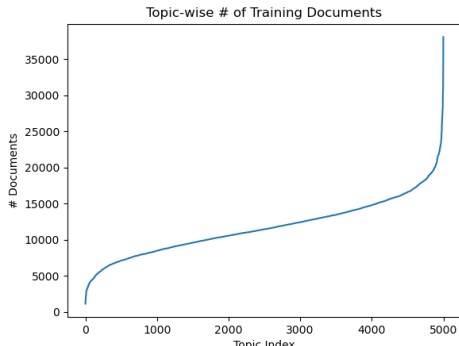

Figure 2: **LoRA-wise perplexity.** Sorted perplexity values of all our trained LoRAs. Underperforming models can be easily identified and further trained for more iterations or with more data if available without affecting the performance of the well-performing ones.

Figure 3: **Number of training documents per topic.** We report the number of documents in the training set of each topic. While there are some outliers with extremely high or low amounts of training data, most topics contain more than 5000 documents.

Table 4: **Text representations of most frequent topics.** In topic-modeling, we cluster the input documents into a number of topics. Here, we provide topic-wise keywords chosen using class based TF-IDF scores for the 15 most frequent topics in the training data. Topics are sorted by their size in descending order. Each cluster seems to be focused on a specific subject and there is diversity in the subject area across clusters.

| Topic Rank | Topic Keywords |
|:---:|:---:|
| 1 | art, museum, artist, artists |
| 2 | orchestra, piano, music, symphony |
| 3 | news, media, journalism, journalists |
| 4 | sports, soccer, football, sport |
| 5 | community, foundation, charity, volunteer |
| 6 | food, foods, nutrition, hunger |
| 7 | stock, shares, market, company |
| 8 | marketing, content, brand, digital |
| 9 | tax, income, taxes, irs |
| 10 | students, university, student, college |
| 11 | fashion, dress, shirt, wear |
| 12 | nuclear, radiation, reactor, weapons |
| 13 | music, musical, jazz, musicians |
| 14 | venezuela, maduro, venezuelan, brazil |
| 15 | sustainability, sustainable, environmental, green |

determine the LoRA to be used at inference. While we provide results with both these approaches in Table 2, note that they are not directly comparable since the number of training tokens are not the same. Tables 4 and 5 present qualitative results on topic modeling. In Table 4, we show the text representations of the ten topics with highest amount of training data. The topic-wise keywords are obtained using class based TF-IDF values of the terms in the topic training data. Highly related keywords within each topic and diverse keywords across topics suggest that the topic modeling is good.

Figure 3 shows the number of training documents per topic for MoIN-5k . We observe that the amount of training tokens is fairly uniform across topics. The topics with very little training data are discarded. It is possible to adapt the number of trainable parameters in the expert (e.g., rank of LoRA) based on the amount of training data for the corresponding topic.

In Table 5, we show sample queries from downstream datasets and the keywords of the topic corresponding to the LoRA selected for the input prompt. We observe that the selected topic is highly relevant to the input for the majority of queries. This demonstrates the effectiveness of our topic modeling and routing mechanisms. The last two rows show examples where the routing mechanism failed and an irrelevant LoRA was chosen.

Table 5: **Topic routing for sample queries.** We provide sample queries from downstream datasets and the keywords of the topic corresponding to the LoRA selected for the input prompt. We observe that the selected topic is highly related to the input for a majority of the queries. The last two rows show examples where the routing mechanism failed and an irrelevant LoRA was chosen.

| Query Prompt | Chosen topic keywords |
|---|---|
| A construction group wants to put a shopping center in town, but the only place available is a small nature park with a trail. Deer and other wildlife frequent the park... | playground, parks, picnic, park, recreation |
| A pot of pasta is boiling on the stove, and the lid on top of the pot is shaking as the water boils more rapidly. A person goes to the stove and removes the pot, releasing steam into the air above, and so the steam is boiling... | pasta, spaghetti, sauce, lasagna, parmesan |
| Personal Care and Style: How to make jewelry cleaner. Select a mild cleaning agent for your homemade jewelry cleaner recipe. Dish soap with grease-cutting properties can be used, but skip dish soap with harsh anti-bacterial properties as this can strip the finish off jewelry surfaces... | grout, stains, stain, vinegar, bleach |
| Neils cat was terrified of thunderstorms but Kyles wasnt bothered by them. Kyle found their cat hiding under the bed after the loud crackle of thunder... | kelsie, cat, yonan, cats, wittle |
| Question: Ethanol is a type of alcohol made from plants. Sugarcane and corn, which are both used in foods such as cereals and breads, are used to make ethanol. Burning ethanol provides a clean source of energy because the products of ethanol are water and... | ethanol, biodiesel, biofuels, biofuel, biomass |
| Hank was eating cereal and spilt milk on his hot pants and decided to get his pleated pants. He needed to change into new leggings because the pleated pants are clean | kaid, kusac, heyes, lijou, rhyaz |
| Question: The instructions below outline the procedure for a demonstration. Materials: four 100 g metal blocks, each of a different metal four polystyrene foam cups, each containing 150 g of 10°C water Procedure: 1. Place the four cups of... | facetentailment, facetid, studentanswer, wa_30b_f5, wa_30b_f1 |

# 5 FUTURE WORKS

The idea of using adapters as experts during pre-training can unlock many new features. While we could only explore this idea in the context of upcycling a model, following are few exciting future directions worth exploring.

**Dynamic parameter count allocation per topic.** Number of training tokens per topic can differ and it may be beneficial to have larger adapters for bigger topics. Hence, LoRA adapters with adaptive rank can be explored to both reduce the total model size and prevent under/overfitting.

**Adapter augmented generation.** Our method shares similarities with retrieval augmented generation (RAG). While RAG retrieves most relevant raw text for a given query, our method retrieves the most relevant adapter. Adapters can be used to memorize and compress the information from several relevant documents. This can make the system more tolerant to errors in retrieval as the adapter stores much more information than raw text.

**Continual/Incremental learning.** Having adapters specialize in different topics makes LLMs modular. This implies that when the underlying data changes, only the adapter corresponding to the changed data needs to be updated. Our framework allows for easier modification of existing experts and integration of new experts without affecting the performance of the rest of the model.

**Quantizing the base model.** Since upcycling involves training the model on a significant amount of data, it should be possible to aggressively quantize the base model similar to QLoRA idea. The experts can recover the lost information during upcycling.

**Better adapters.** Unlike original LoRA adapters which were designed for small-scale fine-tuning, the adapters in our setting ingest much more data. Hence, it is possible to re-visit the adapter architecture to provide better performance/paramter ratio. For instance, inserting an activation function between the two LoRA linear layers may increase its modeling capabilities.

**Creating new experts on the fly.** For queries that fall nicely into one of the topics, our simple routing scheme should work well. However, there may be queries on the boundary of multiple clusters or require access to multiple adapters at the same time. This can be addressed with a parameter server that merges existing adapters based on the complexity of the query.

**Per token adapters.** It is efficient to train the adapters independently in parallel, but once trained, we can explore other costly but better inference strategies where we have per token experts instead of per query. Arrow routing may be useful here (Ostapenko et al., 2024). We can use our technique to bootstrap a library of LoRAs.

**Expert aware pre-training.** Since we're constrained by resources, we could only explore the idea of upcycling an existing model. While less demanding to explore, it could be sub-optimal in that the model has not learnt to offload topic specific information into experts. Hence, expert aware pre-training should be explored where the base model and the experts are trained jointly from scratch.

## 6 CONCLUSION

In this paper, we introduced the mixture-of-introvert experts framework to upcycle a pretrained LLM. The training data is split into semantically related clusters and an expert is trained on each cluster. Queries at test-time are routed to appropriate expert with a simple nearest neighbor search. All the experts can be trained independently. Our approach offers great flexibility in training and provides a way to improve an LLM with limited resources. We provide proof-of-concept experimental results by upcycling a 1B parameter model with 5000 experts on 500B tokens and show comparable or better performance than full-model continued pretraining.

## 7 LIMITATIONS AND BROADER IMPACT

Our approach utilizes large LoRA models, which results in a significant number of parameters compared to standard pretraining methods. Hence, our method is particularly well-suited for applications where multiple instances of the LLM are deployed across several GPUs to serve numerous users, such as in ChatGPT. Moreover, we believe that the size of LoRA models could be substantially reduced; however, we did not explore different ranks for LoRA due to resource constraints. While our method effectively lowers the cost of pretraining LLMs, potentially democratizing the development of novel models, it also raises concerns. Specifically, it may enable less sophisticated adversaries to create their own models, which could lead to negative societal impacts.

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
