# OpenReview forum: "MoIN: Mixture of Introvert Experts to Upcycle an LLM"
_ICLR.cc/2025/Conference — Submitted to ICLR 2025_

### Official Review · Reviewer_EwGL · 2024-10-18

**Soundness:** 2
**Presentation:** 2
**Contribution:** 1
**Rating:** 3
**Confidence:** 5

**Summary:**

This work aims to improve / upcycle existing large language model without the prohibitive requirements of continued pre-training of the full-model. The proposed method split the pre-training data into semantically relevant groups and train an expert (LoRA) on each subset. During inference, the query is routed to the most relevant expert and loaded onto the base model for the forward pass. They claim that this design could empower extreme parallelism without any communication channels between experts.

**Strengths:**

I could hardly figure out the strengths of this work.

**Weaknesses:**

1. Although the main experiment keeps the training token the same for comparison, it would still be an unfair comparison, specifically in terms of the number of training parameters. Suppose LoRA has 1% parameters, 5K LoRAs would be 50 times larger than the base model. It would be unfair to compare the performance with the baselines, given that you have much more parameters.
2. The overall performance of perplexity and downstream tasks might not make senese. While the authors treats pretrained model with less perplexity as better models (tinyllama-3T is the best), there is a sharp decrease on its downstream performance compared to tinyllama-2.5T (~1.5%). Given the performance gain that MoIN-5k has compared to tinyllama-2.5T (0.05%), I hope the authors' further explaination on this.
3. The accessments of the proposed method is insufficient. To be specific, (1) Only one model archtecture is tested. With limited training resources, the authors should conduct more experiments on models like GPT2, OPT. (2) The impact of the number of clusters is not clear. There could be an ablation study.
4. There lacks detailed explanations on the novelty of this work, maybe the authors could add comparisons or dicussions with LoRA MoE.

**Questions:**

The same to weakness.

---

> ### Author Response · Authors · 2024-11-28
> **Response to Reviewer EwGL**
>
> > **1. Although the main experiment keeps the training token the same for comparison, it would still be an unfair comparison, specifically in terms of the number of training parameters. Suppose LoRA has 1% parameters, 5K LoRAs would be 50 times larger than the base model. It would be unfair to compare the performance with the baselines, given that you have much more parameters.**
>
> Yes, we agree that our method contains 50 times more parameters, but that is exactly the goal of our method: trade relatively cheap storage space for reduced resource demands during training. Note that our method does not require all 5000 adapters to be present in the costly GPU RAM all the time since requests get routed at instance (query) level and not token level and adapters can be loaded onto GPU RAM on demand.
>
> > **2. The overall performance of perplexity and downstream tasks might not make senese. While the authors treats pretrained model with less perplexity as better models (tinyllama-3T is the best), there is a sharp decrease on its downstream performance compared to tinyllama-2.5T (~1.5%). Given the performance gain that MoIN-5k has compared to tinyllama-2.5T (0.05%), I hope the authors' further explaination on this.**
>
> Yes, the 3T model seems to be doing worse in terms of downstream accuracy as compared to the 2.T model, but we believe this happens due to saturation during pre-training. The Chinchilla optimal tokens for a 1B parameter model are just 30B. So, we think training it to 3T tokens could be affecting the generalization of the model. This point also supports our method as simply increasing the number of training tokens is not sufficient to improve a model and model parameters must also be scaled. Our method provides an efficient way to do this.
>
> > **3. The accessments of the proposed method is insufficient. To be specific, (1) Only one model archtecture is tested. With limited training resources, the authors should conduct more experiments on models like GPT2, OPT. (2) The impact of the number of clusters is not clear. There could be an ablation study.**
>
> Yes, we agree that experiments with more architectures and cluster sizes would have strengthened our arguments, but we were limited by resources. For instance, the final 500B training run required 40 days on a mix of RTX6000s and 4090s. To run more ablations on cluster sizes and architectures, we would have to forgo this scaling up experiment. Further, LLaMA is an extremely popular architecture and we believe that it would be worth sharing even if the method works only for this architecture.
>
> > **4. There lacks detailed explanations on the novelty of this work, maybe the authors could add comparisons or dicussions with LoRA MoE**
>
> Our method is different from LoRAMoE (https://arxiv.org/abs/2312.09979) in two ways: 1) It uses fine-tuning datasets to train task specific adapters, but we focus on pre-training which doesn’t have task-level grouping of tokens and thus require clustering 2) The adapters are assembled into an MoE layer and a router is used to combine their outputs at the token level. However, in MoIN, routing happens at the instance level and there’s no merging of adapter outputs which drastically simplifies inference and training.

---

> > ### Comment · Reviewer_EwGL · 2024-11-28
> >
> > Thanks for your response.  I have read the rebuttal thoroughly, including the reviews from other reviewers.
> >
> > However, current reponses could hardly address my major concerns.
> >
> > > 1. The authors claimed that this work work aims to **trade relatively cheap storage space for reduced resource demands during training.**
> >
> > I could hardly agree that such implementation is a worthy trade-off, because the usage of 5K LoRAs not only takes more space for storage, but also undermines the model's capability. With such amount of parameters, the model could perform much better than a simple 2B model. Hence, the proposed method also trades off model capability to reduce training resource demands, which the authors seem to take little consideration of.
> >
> > To be more specific, the reduction of resource demands stems majorly from LoRA and 5K LoRAs are designed for the knowledge switch. However, its performance gain in the context of continual pretraining might be marginal from the experiment results, suppose we compare it to simply continual pretraining a larger LM (like 7B) with LoRA.
> >
> > More experiments are required to clarify this concern.
> >
> > > **2. Marginal Improvement Compared to Baselines.**
> >
> > Actually, this point shares some simmilarity with the first concern. The training resource demand reducing comes from LoRA, while 5K LoRA are used for knowledge switch as a form of continual pretraining. In a nutshell, the proposed method uses much more parameters to have LoRA-level resource comsumptions and marginal performance gain. The contribution of the proposed method would not be satisfactory.
> >
> > > **3. Insufficient Experiments.**
> >
> > Many reviewers have raised similar concerns. While I do agree that current experiment setting would cost more time to have more result, in terms of rigor and sufficiency, I would advise the authors to take more detailed and in-depth experiements to make this work prepared as an qualified conference submission.
> >
> > The authors have also claimed that the Chinchilla optimal tokens for a 1B parameter model are just 30B. So what if you reduce the training tokens to provide more ablation results to testify to your viewpoint?
> >
> > > **4. Novelty**
> >
> > 1. While LoRA MoE based methods have task-level grouping tokens, the proposed work simply uses K means clustering via topic modeling. The authors did not treat this part as one of the contributions of this work. More technical novelty explainations on this would be preferred, in writing specifically.
> > 2. The authors propose that the major difference of MoIN and LoRA MoE lies in that MoIN's routing happens at the instance level. In that way, it would be better if the authors could add discussions of retriever-base methods like [1].
> >
> > [1] LoraRetriever: Input-Aware LoRA Retrieval and Composition for Mixed Tasks in the Wild (ACL 2024 finding)

---

> > > ### Comment · Reviewer_EwGL · 2024-11-28
> > > **Cont**
> > >
> > > Since no further revision of this manuscript is submitted and major concerns remain un-solved. I would keep my score and am open to any discussions ongoing.

---

### Official Review · Reviewer_WZkU · 2024-10-18

**Soundness:** 1
**Presentation:** 3
**Contribution:** 1
**Rating:** 3
**Confidence:** 4

**Summary:**

This paper introduces "mixture of introvert experts" (MoIN) for improving LLMs with LoRa adapterss. First they split the training data into semantically relevant groups using topic modeling then train LoRA adapters respectively on each data subset. At inference time, the model routes queries to the most relevant adapter using nearest neighbor search and use it to answer the problem.

**Strengths:**

The authors trained and deployed up to ~5000 independent LoRA adapters, exceeding the number of LoRAs used in previous research.
The inclusion of diagrams and tables with qualitative examples enhances the reader's understanding of the approach.

**Weaknesses:**

Motivation and Method:

1. The paper's efficiency claims primarily stem from the use of LoRA, rather than from any novel contribution of this work.

2. The deployment of thousands of LoRA adapters potentially undermines these efficiency claims, especially in inference stage. A more rigorous analysis comparing the computational requirements of this approach to traditional methods is necessary to substantiate these assertions.

Experimental Design and Results:

3. Building upon TinyLlama-2T instead of training from scratch cannot demonstrate their method's effectiveness across the entire training process.

4. The absence of results for a MoIN-3T model, trained on an equivalent number of tokens as TinyLlama-3T, hinders fair comparison and leaves gaps in understanding the method's scalability.

5. The perplexity comparisons presented in Table 1 are misleading. Achieving lower perplexity on potentially less diverse data subsets doesn't necessarily indicate superior overall learning. The fact that MoIN-5k fails to outperform TinyLlama-3T, which was trained on more diverse data, suggests that the latter may have acquired more comprehensive knowledge.

6. Table 2 reveals that the proposed method doesn't consistently outperform baseline models across various datasets. This inconsistency raises questions about the actual improvements offered by the method and its generalizability.

7. The experiments are confined to a relatively small 1.1B parameter model. Experimental results on more commonly used 7-8B parameter models are expected.

8. The lack of ablation studies makes it challenging to discern the contribution of individual components within the proposed system and how they interact.

**Questions:**

Same as Weaknesses.

---

> ### Comment · Reviewer_WZkU · 2024-11-26
>
> Hi authors, is there any update?

---

> ### Author Response · Authors · 2024-11-28
> **Response to Reviewer WZkU**
>
> > **1. The paper's efficiency claims primarily stem from the use of LoRA, rather than from any novel contribution of this work.**
>
> The novelty of the paper lies in using existing techniques to design an efficient system for continued pre-training. The novelty lies in the way it attacks the problem rather than the solution itself.
>
> > **2. The deployment of thousands of LoRA adapters potentially undermines these efficiency claims, especially in inference stage. A more rigorous analysis comparing the computational requirements of this approach to traditional methods is necessary to substantiate these assertions.**
>
> Similar to most other innovations, our method does not promise free lunch. It’s a way of trading-off increased complexity and resource demands for managing thousands of adapters to have a flexible model that can be up-cycled cheaply. It creates a distributed model where a part resides on fast GPU memory while a part resides on CPU RAM or disk storage and the full model is assembled on the fly for each request on-demand.
>
> > **3. Building upon TinyLlama-2T instead of training from scratch cannot demonstrate their method's effectiveness across the entire training process.**
>
> While it could be applied as a general pre-training method, we do not focus on complete pre-training in this work. We operate in a continued pre-training scenario. Hence, we use TinyLllama-2.5T as the starting point.
>
> >**4. The absence of results for a MoIN-3T model, trained on an equivalent number of tokens as TinyLlama-3T, hinders fair comparison and leaves gaps in understanding the method's scalability.**
>
> We limit the training to 500B tokens due to resource constraints. Thus, we do not report results with a MoIN-3T model.
>
> > **5. The perplexity comparisons presented in Table 1 are misleading. Achieving lower perplexity on potentially less diverse data subsets doesn't necessarily indicate superior overall learning. The fact that MoIN-5k fails to outperform TinyLlama-3T, which was trained on more diverse data, suggests that the latter may have acquired more comprehensive knowledge.**
>
> TinyLlama-3T is trained for 500B tokens more than MoIN-5k so in terms of per-training tokens it is not a fair comparison. Further, 3T model is actually worse than 2.5T version and MoIN-5k (trained for 2.5T tokens in total). We think this is caused by over-training the model since Chinchilla optimal tokens for a 1B parameter model is just 30B.
>
> > **6. Table 2 reveals that the proposed method doesn't consistently outperform baseline models across various datasets. This inconsistency raises questions about the actual improvements offered by the method and its generalizability.**
>
> There are multiple potential reasons for this: 1) The baseline is very strong as it uses the same dataset and same amount of training tokens as ours. 2) We could not optimize the hyper parameters due to resource constraints which could have potentially improved the model performance. 3) Different training recipes can bias the model to be better/worse on some of the downstream tasks.
>
> > **7. The experiments are confined to a relatively small 1.1B parameter model. Experimental results on more commonly used 7-8B parameter models are expected.**
>
> It is not possible for us to train a 7B model with our resources. For instance, the MoIN-5k training run took 40 days on all our GPUs consisting of a mix of RTX6000s and 4090s.
>
> > **8. The lack of ablation studies makes it challenging to discern the contribution of individual components within the proposed system and how they interact.**
>
> Resource constraints limit our options of experiments. We decided to scale-up training from 50B to 500B at the cost of several small-scale ablation studies.

---

> > ### Comment · Reviewer_WZkU · 2024-11-28
> >
> > I have read the author's response. My opinion aligns with Reviewer EwGL's. We have concerns about both the proposed methodology and experiments, and the author's response fails to address our issues. Therefore, I maintain my original score.

---

### Official Review · Reviewer_BY7P · 2024-11-01

**Soundness:** 2
**Presentation:** 3
**Contribution:** 2
**Rating:** 6
**Confidence:** 2

**Summary:**

The authors propose an approach called MoIN, aimed at enhancing an existing large language model without the substantial demands of continued full-model pre-training. To achieve this objective, they suggest clustering the training data into semantically related subsets. Subsequently, they train distinct expert models for each subset and direct queries to the most suitable expert using a straightforward nearest-neighbor search technique. The authors have implemented a proof-of-concept version and conducted experiments on selected models to validate their approach.

**Strengths:**

1. The writing in the paper is clear and accessible, allowing me to easily understand the implementation of the method and the execution of the experiments.

2.  The authors highlight a critical issue within the current MoE framework: since routing varies by token, all experts must reside in GPU memory during both training and inference. This requirement poses challenges for scaling to a large number of experts. I believe this paper could inspire the community to explore alternative strategies for increasing the number of models.

3. The experiments are promising in specific scenarios.

**Weaknesses:**

My main concern is that the experiments conducted are insufficient to support the main conclusions of the paper. The authors only evaluate a limited number of models and datasets. Including more diverse and realistic datasets and models would strengthen the findings and provide a more comprehensive evaluation of the proposed method.

**Questions:**

N/A

---

> ### Author Response · Authors · 2024-11-28
> **Response to Reviewer BY7P**
>
> >**My main concern is that the experiments conducted are insufficient to support the main conclusions of the paper. The authors only evaluate a limited number of models and datasets. Including more diverse and realistic datasets and models would strengthen the findings and provide a more comprehensive evaluation of the proposed method.**
>
> **Experiments with diverse and realistic datasets:**  We train our models on the SlimPajama (Soboleva et al., 2023) which is a deduplicated and cleaned version of the popular open license pretraining dataset RedPajama. We also evaluate on six typical benchmarks for commonsense reasoning.
>
> **Experiments with more models:** We agree that more experiments with different model architectures would strengthen the work. However, we limit our experiments to a single model architecture due to resource constraints.

---

> > ### Comment · Reviewer_BY7P · 2024-11-30
> > **Thanks for the rebuttal**
> >
> > Thanks. I will keep my score.

---

### Official Review · Reviewer_4Vdi · 2024-11-06

**Soundness:** 3
**Presentation:** 3
**Contribution:** 3
**Rating:** 6
**Confidence:** 3

**Summary:**

The paper introduces a novel way to upcycle a pre-trained language model, allowing improvement without the high costs of full model pre-training. The paper utilizes topic models and LoRA adapters to train some independent "introvert" experts to realize more efficient training and inference for the language models.

**Strengths:**

The paper introduces a novel way to upcycle the language models. The new structure of mixed-of-experts allows for efficient parallel inference and training, which provides a framework that could streamline LLM deployments across multiple devices or settings.

**Weaknesses:**

The improvement in language model performance is not obvious to me. In particular, Table 2 shows that MoIN-5k does not consistently outperform TinyLlama-2.5T across most downstream tasks, despite both being trained on the same amount of tokens. This suggests that the main contribution of MoIN may currently lie more in reducing training and inference costs through parallelism rather than in enhancing the model's performance on language tasks.

**Questions:**

1. Out of curiosity, if the goal is to enable the language model to learn from data that falls outside the distribution of the original training set, do the authors have insights on how this might be achieved within the current framework? Additionally, would this approach offer advantages over the continual pre-training of the full model?
2. If the aim is to enhance more abstract qualities of the language model, such as helpfulness, does this framework provide any specific advantages over the original model in achieving such improvements?

---

> ### Author Response · Authors · 2024-11-28
> **Response to Reviewer 4Vdi**
>
> >**The improvement in language model performance is not obvious to me. In particular, Table 2 shows that MoIN-5k does not consistently outperform TinyLlama-2.5T across most downstream tasks, despite both being trained on the same amount of tokens. This suggests that the main contribution of MoIN may currently lie more in reducing training and inference costs through parallelism rather than in enhancing the model's performance on language tasks.**
>
> Yes, that is correct. Our main contribution is in reducing the training cost while keeping the inference cost approximately the same as the full model through parallelism. We strongly believe that exploring this direction more and improving the training recipe can also improve the model accuracy.
>
> > **1. Out of curiosity, if the goal is to enable the language model to learn from data that falls outside the distribution of the original training set, do the authors have insights on how this might be achieved within the current framework? Additionally, would this approach offer advantages over the continual pre-training of the full model?**
>
> That is an excellent question! We don’t experiment with out-of-distribution training dataset (we sample a 500B subset from the original pre-training dataset). However, we can extend our method to train a subset of adapters on each different data distribution, and route an incoming request by first classifying it as one of the distributions. This is exactly the sort of application we had in mind when working on this method.
>
> >**2. If the aim is to enhance more abstract qualities of the language model, such as helpfulness, does this framework provide any specific advantages over the original model in achieving such improvements?**
>
> It may be necessary to fine-tune the base model for improving its abstract qualities like helpfulness which can reduce the benefits of our framework.

---

### Meta-Review · Area_Chair_G3Fa · 2024-12-19

**Metareview:**

The paper presents an alternative to the building of a single LLM, through designing a host of light LLMs dedicated to different topics; each ,light LLM is defined as a LoRA adapter, and queries are routed to the appropriate expert.
The strategy can be summarized as "creating a distributed model where a part resides on fast GPU memory while a part resides on CPU RAM or disk storage". The full model is assembled on the fly for each request on-demand thus trading relatively cheap storage space for reduced resource demands during training.

**Additional Comments On Reviewer Discussion:**

The LLM field notoriously goes with heavy experiments. The reviewers consider that the presented experiments are insufficient to support the paper claims. The authors reply that resource constraints limit their experiments, especially in the imparted time.

The key discussion is about the trade-off between memory resources/ training time/ performances.
After reviewer EwGL,  "5K LoRAs not only takes more space for storage [than a standard large LLM], but also undermines the model's capability. With such amount of parameters, the model could perform much better than a simple 2B".

The area chair appreciates the goal (building more frugal LLMs based on ensemble approaches); however, the claims in this field must be validated through heavy experiments.

---

### Decision · Program_Chairs · 2025-01-22

Reject